# Systemic Delivery of Magnetogene Nanoparticle Vector for Gene Expression in Hypoxic Tumors

**DOI:** 10.3390/pharmaceutics15092232

**Published:** 2023-08-29

**Authors:** Luis Daniel Terrazas-Armendáriz, Cynthia Aracely Alvizo-Báez, Itza Eloisa Luna-Cruz, Becky Annette Hernández-González, Ashanti Concepción Uscanga-Palomeque, Mitchel Abraham Ruiz-Robles, Eduardo Gerardo Pérez Tijerina, Cristina Rodríguez-Padilla, Reyes Tamez-Guerra, Juan Manuel Alcocer-González

**Affiliations:** 1Laboratorio de Inmunología y Virología, Facultad de Ciencias Biológicas, Universidad Autónoma de Nuevo León, San Nicolás de los Garza 66450, NL, Mexico; luis.terrazasarmn@uanl.edu.mx (L.D.T.-A.); cynthia.alvizobz@uanl.edu.mx (C.A.A.-B.); itza.lunacrz@uanl.edu.mx (I.E.L.-C.); becky.hernandezgnz@uanl.edu.mx (B.A.H.-G.); ashanti.uscangapl@uanl.edu.mx (A.C.U.-P.); crrodrig07@gmail.com (C.R.-P.); rtamez1804@yahoo.com.mx (R.T.-G.); 2Centro de Investigación en Ciencias Fisico Matematicas, Facultad de Ciencias Físico Matematicas, Universidad Autónoma de Nuevo León, Ciudad Universitaria, San Nicolás de los Garza 66451, NL, Mexico; mitchel.ruizrb@uanl.edu.mx (M.A.R.-R.); eduardo.pereztj@uanl.edu.mx (E.G.P.T.)

**Keywords:** Magnetogene, magnetic nanoparticles, magnetofection, hypoxia, chitosan, gene expression

## Abstract

Cancer is a disease that causes millions of deaths per year worldwide because conventional treatments have disadvantages such as unspecific tumor selectivity and unwanted toxicity. Most human solid tumors present hypoxic microenvironments and this promotes multidrug resistance. In this study, we present “Magnetogene nanoparticle vector” which takes advantage of the hypoxic microenvironment of solid tumors to increase selective gene expression in tumor cells and reduce unwanted toxicity in healthy cells; this vector was guided by a magnet to the tumor tissue. Magnetic nanoparticles (MNPs), chitosan (CS), and the pHRE-Luc plasmid with a hypoxia-inducible promoter were used to synthesize the vector called “Magnetogene nanoparticles” by ionic gelation. The hypoxic functionality of Magnetogene vector nanoparticles was confirmed in the B16F10 cell line by measuring the expression of the luciferase reporter gene under hypoxic and normoxic conditions. Also, the efficiency of the Magnetogene vector was confirmed in vivo. Magnetogene was administered by intravenous injection (IV) in the tail vein and directed through an external magnetic field at the site of tumor growth in C57Bl/6 mice. A Magnetogene vector with a size of 50 to 70 nm was directed and retained at the tumor area and gene expression was higher at the tumor site than in the others tissues, confirming the selectivity of this vector towards hypoxic tumor areas. This nanosystem, that we called the “Magnetogene vector” for systemic delivery and specific gene expression in hypoxic tumors controlled by an external magnetic designed to target hypoxic regions of tumors, can be used for cancer-specific gene therapies.

## 1. Introduction

Cancer is currently one of the major public health challenges and is the second leading cause of death worldwide. Current treatments are limited by their therapeutic inefficiency due to lack of tumor selectivity, high therapeutic doses, and side effects. For this reason, alternative developments are being sought for new cancer gene therapies that can be targeted and concentrated in tumor cells, controlling the release of drugs or genetic material and thus decreasing the side effects in healthy tissues [1]. One important feature of human solid tumors is their low oxygen level which generates hypoxia. Intratumoral hypoxia has been considered a force leading to tumor progression with negative prognosis in patients [2,3]. As tumors develop regions of hypoxia, they must adjust their metabolism to adapt to this oxygen-depleted microenvironment. Tumors acclimate through the activation of hypoxia inducible factor (HIF), which plays an essential role in the switch to anaerobic energy production [4]. The HIF-1 transcription factor is the major regulator of tumor adaptation to hypoxia and induces the expression of many genes that participate in angiogenesis, the metabolism of iron and glucose, cell survival and proliferation [5,6], and allow cells to survive in these conditions [7]. The hypoxia-responsive element (HRE) is the minimal indirect *cis*-regulatory element transactivated by the hypoxia-inducible factor (HIF). Data from over 70 genes suggest that endogenous HREs are composite regulatory elements comprising the conserved HIF binding site (HBS) with an A/GCGTG core sequence and a highly variable flanking sequence. A single HBS is essential but not sufficient for activation under hypoxic conditions [8]. The binding sites of transcription factors provided by the flanking sequences are not necessary for hypoxia regulation but are required to amplify the hypoxic response or make the HRE tissue-specific [9]. The HRE/HIF1 regulatory system is common in all mammalian cells and human tissues and can be used to achieve the selective expression of therapeutic genes under hypoxic conditions [10]. When HREs derived from different genes are placed in plasmid vector systems, they confer hypoxic inducibility on heterologous promoters in various cell types [11]. The cells of hypoxic regions in solid tumors are chemotherapy-resistant due to the limited delivery of drugs to those regions via the circulation [12,13]. The lack of an efficient transport system in the body for delivering drugs to tumor cells has recently attracted substantial attention, and many delivery vehicles have been postulated. Magnetic nanoparticles (MNPs) have attracted increasing attention for their excellent physicochemical properties and promising applications such as in drug delivery, magnetic transfection, and cancer therapy [14,15]. MNPs are an excellent material in nanomedicine due to their superparamagnetism, magneto-thermal effect, and ability to manipulate particle movement [15,16]. However, it is important to mention that the oxidation and acid erosion of the surface of MNPs is an important problem since their degradation is more rapid. In addition, MNPs must interact with biological molecules and evade the reticuloendothelial system (RES) [16]. Therefore, these complexes are coated with polymers such as starch, dextran, polyethylene glycol (PEG), and chitosan, thus increasing their circulation time, biocompatibility, biodegradability, and hydrophobicity and increasing the probability of them reaching their target cells. Among these polymers, chitosan is an attractive drug or gene delivery system [17,18]. Magnetic drug delivery systems work via the delivery of magnetic nanoparticles loaded with the drug to the tumor site under the influence of an external magnetic field [19,20]. In vivo magnetic fields are focused on the required site to promote transfection and to deliver the therapeutic gene to a specific organ or site within the body. Many different tissues have been studied as potential magnetic drug delivery targets [20,21]. The main objective of this study was to construct a vector for selective gene expression in the hypoxic microenvironments of mouse solid tumors through the development of the Magnetogene nanoparticle vector. This system is based on three components: (1) MNPs—by applying a magnetic field these nanoparticles can be directed and focused; (2) pHRE-Luc plasmid—it contains a promoter that overexpresses a gene under hypoxia conditions which provides selective gene expression; and (3) chitosan—this cationic polymer acts as a protective agent, provides anchoring links, and controls the release of the pHRE-Luc plasmid, in addition to giving greater biocompatibility to this gene delivery system.

## 2. Material and Methods

### 2.1. Chemicals

Water soluble 20 kDa chitosan, with a degree of deacetylation of ~70%, from Coyote Foods (Saltillo, CH, Mexico) and Pentasodium Tripolyphosphate (TPP) from Sigma–Aldrich (St. Louis, MO, USA) were weighed and resuspended in MiliQ water, at concentrations of 2 mg/mL and 0.86 mg/mL, respectively. The pH of chitosan was measured and adjusted to 5.5, while for TPP it was adjusted to 3. Cell culture media, fetal bovine serum (FBS), cell culture supplements, and plasmid purification kit were obtained from Invitrogen (Carlsbad, CAL, USA). Iron II sulfate heptahydrate and iron III chloride hexahydrate were purchased from Sigma–Aldrich (St. Louis, MO, USA), ammonium hydroxide was obtained from Productos Quimicos Monterrey (Monterrey, NL, Mexico).

### 2.2. Plasmids

Two plasmids were used for this study: the first was pHRE-Luc, it contains the luciferase gene regulated by a hypoxia-inducible promoter. This was generated by 6 consensus sequence repeats called hypoxia response elements (HRE), based on of HIF1α and HIF2α of the EPO and vascular endothelial growth factor (VEGF) genes (CGTGTACGTG) separated by 5 random nucleotides, followed by the minimal promoter of human thymidine kinase (TK). These six tandem repeats of HRE interact with the transcription factors HIF1α and HIF2α. The second was pRNAi interference of HIF 1 alpha (pGSH1-GFP) which inhibits the expression of the HIF 1 alpha gene. The vector pGSH1-GFP was used to test the hypoxic gene inducibility of pHRE-Luc.

### 2.3. In Vitro Studies

To determine the selective hypoxic expression of pHRE-Luc vector, we used the B16F10 cell line (ATCC^®^ CRL6322 TM) of mouse melanoma. Cells were grown in Dulbecco’s Modified Eagle’s medium (DMEM) supplemented with 10% FBS and antimitotic 1% antibiotic (Invitrogen, Thermo Scientific, Waltham, MA, USA). A total of 3000 cells per well were seeded in 96-well plates. Cells were incubated until 80% confluence was reached. Polyethylenimine (PEI) (4.3 mg/mL) was used as the transfection agent. The plasmids pHRE-Luc (1 µg) and pGSH1-GFP (1 µg) were transfected through the preparation of PEI-pDNA complexes and cells were cultured at 37 °C in hypoxic and normoxic conditions for 24 h. All samples were performed in triplicate. To determine the expression of the reporter gene, luciferase induction by the HREs attached to the promoter was measured by adding luciferin substrate to the cell lysate. For this assay we used the Steady-Glo Luciferase kit assay system from Promega. After 24 h, the medium was removed from the wells, and the cells were washed with sterile PBS, 100 μL of lysis buffer was added to each well, the plate was shaken to homogenize the material, and then it was incubated for 10 min at RT. Then, 100 μL of the redissolved luciferin-containing reagent was added to each tube, and the luciferase was finally quantified in Varioskan Lux (Thermo Scientific, Waltham, MA, USA).

### 2.4. Magnetic Nanoparticles (MNPs) Synthesis and Characterization

Iron oxide magnetic nanoparticles were prepared using a minor modification of a protocol reported by Bae et al. (2012) [22]. An amount of 202.6 mg of FeO_4_S·7H_2_O and 546.1 mg of Cl_3_Fe·6H_2_O were dissolved in 100 mL of deionized water (ddH2O); once dissolved, 12 mL of NH_4_OH was added by dripping to a three-necked flask with magnetic stirring at 60 °C along with a constant flow of argon to maintain an oxygen-free environment. Synthesis of MNPs was incubated for 1 h at 950 rpm and 60 °C. Subsequently, magnetic nuclei were recovered by means of an external magnetic field, and washed with ddH2O, methanol, and acetone sequentially, which was repeated 20 times. MNPs size and morphology were determined using transmission electron microscopy (TEM). A drop of the aqueous MNPs dispersion was added to a coated copper grid, subsequently dried with a vacuum desiccator, and then examined under a transmission electron microscope. Histograms with size distribution were made in software image J [23].

### 2.5. Magnetogene Synthesis and Characterization

The Magnetogene nanoparticle vector is composed of MNPs, pHRE plasmid, and chitosan to protect and encapsulate the pDNA. Magnetogene was prepared using ionotropic gelation between positively charged chitosan and negatively charged TPP, as first reported by Calvo et al. (1997) [24], 750 µL of chitosan (2 mg/mL) and 750 µL of PBS was added to the tube labeled as positive, while 5000 ng of MNPs, 50 µg of pHRE-Luc, 187.5 µL of TPP, and 1237.5 µL of PBS were added to the tube labeled as negative. The contents of the negative tube was added dropwise to the positive tube under constant shaking; then, they were incubated for 1 h at room temperature with shaking of 950 rpm. The final contents was adjusted to a volume of 100 µL. Size and zeta potential of Magnetogene were determined by dynamic light scattering (DLS) using the Zetasizer apparatus (ZS90, Malvern. UK) and Zetasizer Software version 7.11. The refractive index (RI) and absorption used for Magnetogene measurements were 1.330 and 0.001, respectively (chitosan values), while for the dispersant (water) the following values were used: RI of 1.330, a viscosity of 0.8872 cP, and a dielectric constant of 78.5. The results obtained from the three measurements are reported in a size and zeta-potential graph. Scanning electron microscopy (SEM) analysis was performed using the JOEL model JSM-6390LV (Jeol USA Inc., Peabody, MA, USA) at a working voltage of 20 kV and a magnification of 55,000×. Amounts of 20 µL of Magnetogene nanoparticle vector and 20 µL of the 1% fixative solution (0.1 M phosphate buffer and 25% glutaraldehyde) were added on a coverslip and allowed to fix for 30 min at 4 °C. The remaining liquid was removed and dried to visualize the sample in SEM.

### 2.6. In Vivo Studies

For in vivo studies, female C57BL/6 mice (6–8 weeks old) from Harlan Laboratories (Indianapolis, IN, EU) were used. Mice were injected with 1 *×* 10^6^ B16F10 melanoma cells intramuscularly in the thigh, and solid tumor growth was observed in the first 20 days. Mice were divided into 4 groups of 3 mice: 1 Negative control, 2 Magnetogene + magnet, 3 Magnetogene without magnet, and 4 Magnetogene with pGSH1-GFP + magnet. All Magnetogene nanoparticles vector groups (5000 ng of MNPs, 50 µg of pDNA) were injected to the mice through the tail vein and a magnet was placed on the thigh of the tumor bearing mice for 1 h. Mice were sacrificed 48 h post injection for analysis. The organs collected (heart, kidney, lung, liver, spleen, and tumor tissue) were homogenized with PBS and 100 µL of lysis reagent buffer. Then, 100 µL of supernatant of the lysate of each organ were added in white flat-bottom 96-well plates from Costar Corning (Corning, NY, USA). To analyze the luciferase activity, the ONE-Glo Luciferase assay system kit from Promega (Madison, WI, USA) was used according to the manufacturer’s recommendations. A total of 100 µL of the lysate was used and analyzed by a Varioskan Lux luminometer from ThermoFisher Scientific. All animal-handling procedures were performed according to the Mexican Official Standard NOM-062-ZOO-1999, technical specifications for the production, care, and use of laboratory animals. This study was approved by the Comité de Ética de Investigación de Bienestar Animal of the Biological Sciences Faculty (CEIBA) of the Autonomous University of Nuevo Leon (UANL). All animal-handling procedures were performed according to the Mexican Official Standard’s NOM-062-ZOO-1999 technical specifications for the production, care, and use of laboratory animals.

### 2.7. Statistical Analysis

Statistical analysis of the data was performed using the GraphPad Prism version 6.0 software, where the normality of the data was verified and, subsequently, a post ANOVA test was performed for the multiple comparison of means using the Tukey test.

## 3. Results

### 3.1. Selective Hypoxic Expression of pHRE-Luc Vector in B16F10 Cells

The hypoxic inducibility of pHRE-Luc plasmid was demonstrated by transfecting the vectors pGSH1-GFP and pHRE-Luc naked into the B16F10 cells under normoxic and hypoxic conditions. The levels of luciferase on the pHRE-Luc vector under the hypoxic condition 1,615,768 RLUs (*p* < 0.0001) were significantly higher that the normoxic condition 379,851 RLUs. Also, to evaluate the luciferase gene activation, tumor hypoxia cells were transfected with pGSH1-GFP in hypoxia environments. It was observed that luciferase levels were reduced significantly in the pHRE-Luc/pGSH1-GFP group (55,939 RLUs) compared with the pHRE-Luc alone group (1,615,768 RLUs) (*p* < 0.0001) (Figure 1).

### 3.2. Magnetogene Synthesis and Characterization

Magnetogene has three components: MNPs, chitosan, and pHRE-Luc vector. MNPs were first synthesized.

MNPs were synthesized using the coprecipitation method and characterized by TEM. Figure 2 shows that the morphology of the MNPs is homogeneous and most are spherical. The size of the MNPs determined from a TEM micrograph are between 10 and 16 nm. Histograms with size distribution were made in software image J.

After MNPs were synthesized, Magnetogene was prepared using the method described above.

The DLS-measured hydrodynamic size distribution of Magnetogene showed complete homogeneity, a low polydispersity index of 0.174, a size of 218 nm with 100% intensity, and St Dev of 61.3 (Figure 3C). The zeta potential was +15.9 mV with an intensity of 100% and St Dev of 5.15 (Figure 3C). On the other hand, Magnetogene was characterized by SEM. Figure 3A confirms the homogeneity and sphericity of the Magnetogene (60 nm). As seen in the SEM, no agglomerations are observed here (Figure 3A).

### 3.3. Measurements of Magnetogene Levels in Mice Tissue

We determined the Magnetogene levels in mice tissue. The Magnetogene vector was developed from pHRE-Luc or pGSH1-GFP plasmids, chitosan, and MNPs, then Magnetogene vectors were administered intravenously in tumor-bearing mice. To direct the vector towards the tumor area, a magnet was placed for 1 h (Figure 4). After 48 h, luciferase expression levels were determined in the following organs: heart, kidney, lung, liver, spleen, and tumor tissue.

The efficiency of the expression of the luciferase reporter gene in the different organs (spleen, heart, liver, lung, kidney, and tumor tissue) was analyzed using the luminescence intensity. The group with Magnetogene with the influence of a magnetic field (M + M) had a greater intensity of light in the tumor tissue (6631 units of light (RLU)/mg protein) compared to that with the group of Magnetogene without the influence of a magnetic field (M-M) (2277 RLU/mg protein) (*p* < 0.0001) and the negative control (849 RLU/mg protein) (*p* < 0.0001). In addition, a decrease in light intensity was observed in the with Magnetogene + pGSH1-GFP with the influence of a magnetic field (M + H + M) (4868.1413 UL/mg protein) (*p* < 0.01) which contains an iRNA directed against HIF (Figure 5). Tumor tissue had the highest light intensity compared to the other organs, indicating that the Magnetogene nanoparticle vector was able to target the tumor area after intravenous administration in tumor-bearing mice and release its content (pHRE-Luc).

## 4. Discussion

Current cancer therapies lack tumor selectivity, which is why tumor gene therapies seek greater therapeutic efficiency and reduce unwanted side effects. The main objective of this study was to construct a Magnetogene nanoparticle vector for selective gene expression in the hypoxic microenvironments of mouse solid tumors. In the present investigation, a novel hypoxic vector was used (pHRE-Luc) in which the luciferase expression was regulated by a eukaryotic promoter containing six tandem repeats (CGTGTACGTG) of the minimal binding sites of HIF1α and HIF2α. Hypoxia response elements (HRE) are regulatory consensus sequences comprising a highly conserved HBS. Under the conditions of hypoxia, HIF-1 binds to the consensus DNA sequences of 5′-(A/G) CGTG-3′ HREs found in promoter regions, which are regulated by HIF-1a. Several studies carried out by other researchers indicate that the hypoxic environment can be used to activate the heterologous gene expression mediated by HRE [25,26]. In the present investigation, the efficiency of the pHRE-Luc vector was demonstrated through in vitro assays involving transfections with B16F10 cells under hypoxic conditions, which contained the vector of the six replicates in tandem and showed a greater ability to regulate luciferase expression than that of cells under conditions of normoxia with a *p* < 0.0001 (Figure 1).

Similar to our results, Dachs et al. (1997) found that an HRE of the mouse gene fofoglicerate kinase 1 can be used to control the expression of a marker and therapeutic genes in vitro and in vivo [27]. In another study, Shibata et al. (2000) developed a vector that responds to hypoxia, which contained 5XHRE derived from the VEGF gene combined with the minimal cytomegalovirus (CMV) promoter. Cells transfected with the vector showed a significant increase in gene expression under hypoxic conditions; the levels were similar to the level of the intact CMV promoter [12,27]. In addition, it was shown, by an iRNA directed against HIF, that the activation of the hypoxic promoter of the pHRE-Luc vector is dependent on the interaction of the HIF1α, HIF2α, and HRE proteins; the use of a cell line transfected with the pHRE-Luc vector was observed under hypoxic conditions containing the iRNA, and significantly decreased luciferase expression was observed compared to those that did not contain the iRNA. Under low oxygen conditions, PHD/VHL is inactivated, which is involved in the degradation of HIF and resulted in an accumulation of protein. Similar results were seen by Zhou and coworkers, who created a reporter vector of HIF (6HRE-GFP) that responded to the accumulation of this protein; they tested an iRNA directed against HIF1α-HIF2 α and observed that the intensity of GFP noticeably decreased. Their 6HRE-GFP vector contained six repeats of HRE as well as the minimal promoter of TK and GFP and, by transfecting the vector in different cell lines under hypoxic conditions, significant increases in the expression of the heterologous gene were observed; similar increases were observed in this study. The number of copies of HRE is an important component in the regulation of the expression of genes induced by hypoxia [26].

To prove the efficiency of our vector, it was encapsulated in chitosan magnetic nanoparticles, called Magnetogene, with a final size of 60 nm by SEM and 218 nm by DLS (Figure 2). DLS requires the sample to be dispersed in solution, while for SEM, a dry flat surface sample is used, and the difference in the sample can influence the morphology of the Magnetogene due to the flexible nature of the chitosan polymer [20,28]. Magnetogene has a Zeta potential of +15.9 mV which is a critical parameter for cellular interaction; interactions with pDNA-NP complexes indicate that the complexes were positively charged. Nanoparticles with higher surface charges bind more strongly to the cell membrane and show a higher cellular uptake because electrostatic interactions between the anionic membrane and cationic nanoparticles facilitate uptake [29,30]. After the adsorption of the nanoparticles onto the cellular membrane, the uptake occurs via several possible mechanisms, such as pinocytosis, nonspecific or receptor-mediated endocytosis, or phagocytosis [31,32].

Once characterized, the Magnetogene vector was inoculated via IV into the caudal vein of mice, then, 48 h post inoculation, the expression efficiency of the reporter gene in the Magnetogene vector was determined. The highest intensity of luminescence was detected in the tumor tissue compared to the other organs (spleen, heart, liver, lung, and kidney). This is because the hypoxic regions are characteristic of the tumor regions, which generates the activation of HIF [33]. This transcription factor recognizes the tandem repeats of the minimum binding sites of HIF1α and HIF2α placed in the promoter of the pHRE-Luc vector used in this study, which generated a high expression of the luciferase gene in the tumor tissue. The level of luciferase expression is dependent on the levels of HIF in the tissues. This was demonstrated by an iRNA directed against HIF in the group where the iRNA was used; the luminescence intensity level decreased significantly. Zhou et al. developed a vector with a similar promoter. When transfecting cells in a hypoxic environment they observed that the intensity of eYFP was significantly higher than in cells with normal oxygen levels; in the same way they observed that, when a siRNA was transfected against HIF, the level of eYFP decreased [26]. In this study, we also observed that, in the group where there was an absence of a magnetic field to direct the Magnetogene vector, the luminescence signal decreased three times compared to the group that did have the presence of a field (Figure 5). When a magnetic field was not applied, the expression of the reporter gene of Magnetogene was not specific to a target organ. This result indicates that, when a magnetic field is placed in a specific site, it is capable of directing the Magnetogene vector through the venous circulation to a specific, desired organ or tissue. In 2016, nanoparticles with chitosan and pDNA were inoculated via IV in C57/Bl6 mice. Using a magnet, they directed the nanoparticles to the tumor foci in the lungs of the mice where there was a greater expression of the apoptotic gene loaded on the nanoparticles [34].

These results are consistent with our previous investigations in which we used an in vivo magnetofection technique. The results showed that the magnetic field could direct the Magnetogene to the tumor site where the field was placed, the gene expression of luciferase could be controlled, and the promoter could be activated under the hypoxic conditions of the tumor.

## 5. Conclusions

○The major component of the Magnetogene nanoparticle vector (pHRE-Luc) had increased gene expression under hypoxia conditions; this was demonstrated in B16F10 cells.○Magnetogene vector nanoparticles with a size of ~60 nm were synthesized.○The Magnetogene nanoparticle vector can be targeted and concentrated in tumor cells by applying a magnetic field, since a higher concentration of this vector was obtained in the tumor tissue of mice after intravenous inoculation into the tail of the mice.○The Magnetogene nanoparticle vector has a higher selective gene expression in hypoxic tumor cells than in healthy cells and tissues.

The Magnetogene nanoparticle vector has important characteristics that respond to an external magnetic field and selective gene expression in hypoxic tissue. These two qualities make Magnetogene an ideal vector for specific solid tumor therapies.

## Figures and Tables

**Figure 1 pharmaceutics-15-02232-f001:**
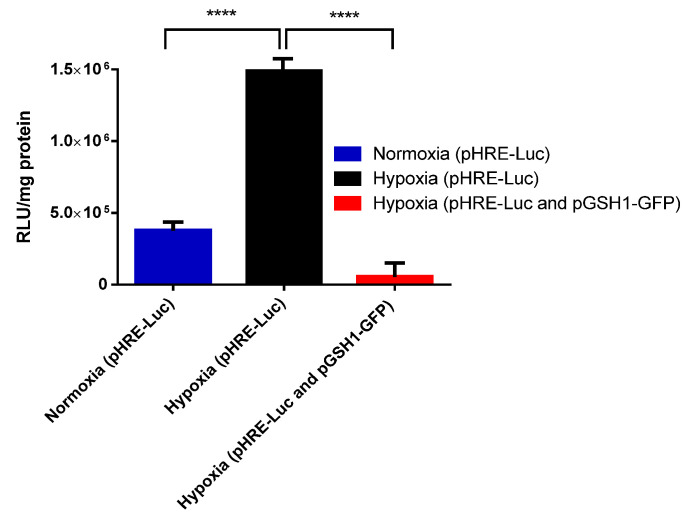
Luciferase activity in B16F10 cells transfected with pHRE-Luc and pGSH1-GFP plasmids under normoxic and hypoxic conditions. Tukey’s test was used to analyze the difference between the means. The data represent the mean ± the standard error of the average of 3 repetitions. The statistical difference is shown (**** *p* < 0.0001).

**Figure 2 pharmaceutics-15-02232-f002:**
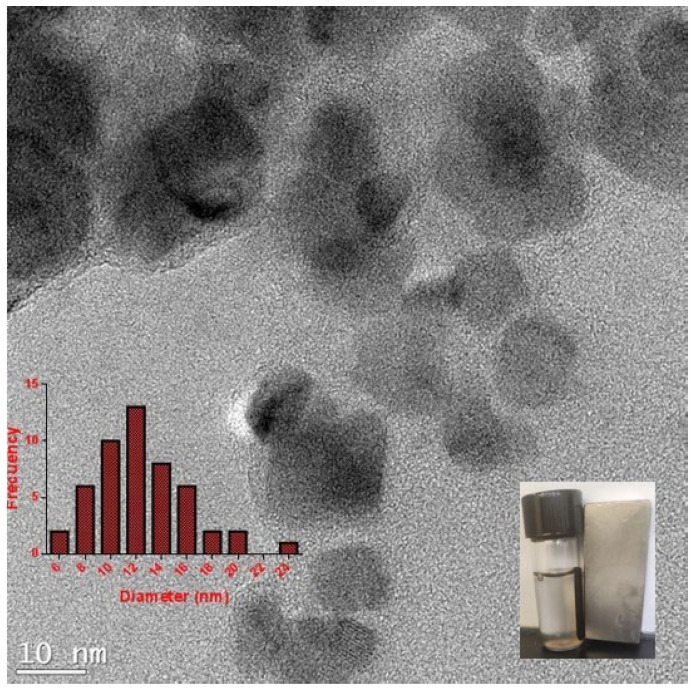
Transmission electron micrograph of synthesized magnetic nanoparticles.

**Figure 3 pharmaceutics-15-02232-f003:**
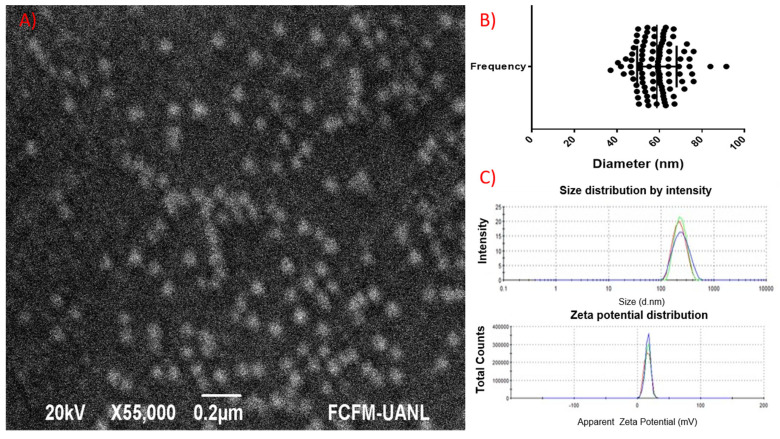
(**A**) Scanning electron micrograph of Magnetogene vector. (**B**) Frequency distribution data. Size and (**C**) zeta potential of Magnetogene vector.

**Figure 4 pharmaceutics-15-02232-f004:**
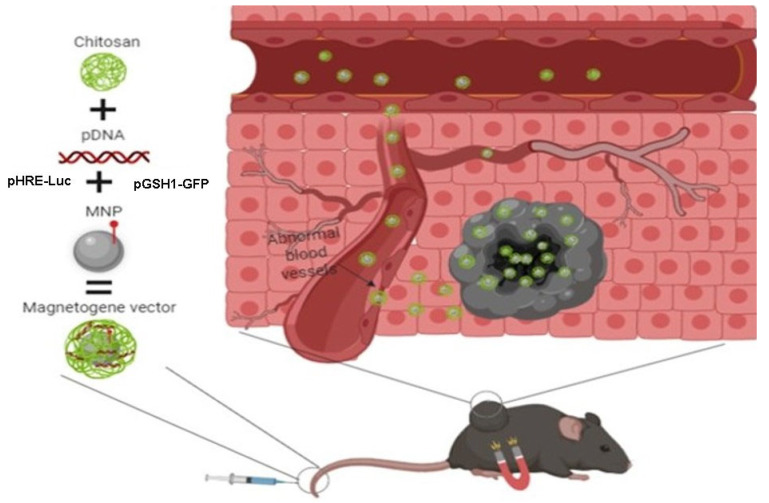
Mice magnetofection with the Magnetogene nanoparticle vector. Created with BioRender.com.

**Figure 5 pharmaceutics-15-02232-f005:**
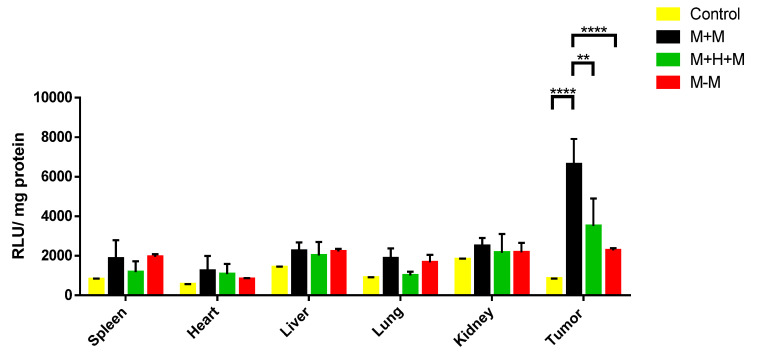
Luciferase gene expression in different organs of mice. Notes: Relative luminescence intensities plotted in a bar graph displaying signals for different organs of tumor-bearing mice 48 h after IV of Magnetogene nanoparticle vector. RLU/mg protein was normalized to milligrams of protein. All treatments have the vector pHRE-Luc. The Tukey’s test was used to analyze the difference between the means. The data represent the mean ± the standard error of the average of 3 repetitions. The statistical difference is shown (** *p* < 0.01, **** *p* < 0.0001).

## Data Availability

The data used to support the findings of this study are included within the article.

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
