# Peer review of "Systemic Delivery of Magnetogene Nanoparticle Vector for Gene Expression in Hypoxic Tumors"

_pharmaceutics, 2023, doi:10.3390/pharmaceutics15092232_

Round 1
Reviewer 1 Report
1. What is the purpose of the author comparing the pHRE-Luc in hypoxia vs PHRE-Luc and PGSH1-GFP in hypoxia in Figure 1? The author should provide more comments on that.
2. In Figure 1, the author should provide the normoxia PHRE-Luc and PGSH1-GFP as control too.
3. The author did the in vivo injection using PHRE-Luc plasmid, chitosan and MNPs, what is the purpose using chitosan and MNPs? Why don’t the author just inject PHRE-Luc plasmid?
4. The author should evaluate the in vitro transfection of those three mixing on B16-F10 cells
5. What is the in vitro cytotoxicity and in vivo tolerability of the PHRE-Luc plasmid, chitosan and MNPs? The author should provide those data too.
6. I would recommend the author use the IVIS to provide straightforward way to see the FLuc expression across different organs in Figure 5.
English is fine to understand.
Author Response
Response to reviewer 1
Comments and Suggestions for Authors
- What is the purpose of the author comparing the pHRE-Luc in hypoxia vs PHRE-Luc and PGSH1-GFP in hypoxia in Figure 1? The author should provide more comments on that.
Response to reviewer
The purpose of this comparison is to demonstrate that pHRE-Luc expression is higher under hypoxia than under normal oxygen conditions because under hypoxia there is an upregulation of the transcription factor HIF. Using pGSH1-GFP it was shown that pHRE-Luc is dependent on HIF accumulation since pGSH1-GFP is an RNA interference that inhibits HIF expression. This information is described in the plasmids description, results and discussions.
- In Figure 1, the author should provide the normoxia PHRE-Luc and PGSH1-GFP as control too.
Response to reviewer
Figure 1 is intended to show 2 comparisons: pHRE-Luc under normoxia and hypoxia conditions to show that pHRE-Luc vector has higher luciferase expression under hypoxia conditions. The other comparison was of pHRE-Luc Vs pGSH1-GFP only under hypoxia conditions this showed that when HIF transcription factor expression is inhibited luciferase expression decreases because pHRE-Luc is dependent on HIF accumulation and this only happens under hypoxia conditions.
- The author did the in vivo injection using PHRE-Luc plasmid, chitosan and MNPs, what is the purpose using chitosan and MNPs? Why don’t the author just inject PHRE-Luc plasmid?
Response to reviewer
The Magnetogene Nanoparticle vector has 3 components: pHRE-Luc, chitosan and MNPs. The function of each of the components is explained in the final part of the introduction (was incorporated for this review) and reads as follows:
1; MNPs; by applying a magnetic field these nanoparticles can be directed and focused. 2; pHRE-Luc plasmid; it contains a promoter that overexpresses a gene under hypoxia conditions which provides selective gene expression. 3; Chitosan; this cationic polymer acts as a protective agent, provides anchoring links and controls the release of the pHRE-Luc plasmid, in addition to giving greater biocompatibility to this gene delivery system.
- The author should evaluate the in vitro transfection of those three mixing on B16-F10 cells
Response to reviewer
It was decided not to use this test because our team has already determined the CC50 of these three components in an unpublished work derived from a master's thesis (Carolina Bonilla Medina. 2017. Evaluación de microbots basados en eritrocitos y nanopartículas magnéticas para el envío dirigido de genes a tumores. Master's thesis. UANL. eprints.uanl.mx/14227) the concentrations we used in this study are below the CC50 determined.
- What is the in vitro cytotoxicity and in vivo tolerability of the PHRE-Luc plasmid, chitosan and MNPs? The author should provide those data too.
Response to reviewer
- It was decided not to use this test because our team has already determined the CC50 of these three components in an unpublished work derived from a master's thesis (Carolina Bonilla Medina. Evaluación de microbots basados en eritrocitos y nanopartículas magnéticas para el envío dirigido de genes a tumores. Master's thesis. UANL. eprints.uanl.mx/14227) the concentrations we used in this study are below the CC50 determined.
- I would recommend the author use the IVIS to provide straightforward way to see the FLuc expression across different organs in Figure 5.
Response to reviewer
Without a doubt it would be a very valuable test that would contribute a lot to this work however the main objective was demonstrated with the quantification of FLuc from the organs removed from the mice and it was demonstrated that the Magnetogene vector can be guided and concentrated in the hypoxic tissue of the tumors of the mice in addition to generating a hypoxic gene inducibility in the tumor tissue.

Reviewer 2 Report
The manuscript presents a novel approach for cancer-specific gene therapy using the Magnetogene vector. The authors successfully demonstrate its hypoxic inducibility and targeted delivery in tumor-bearing mice. However, the paper would benefit from addressing the mentioned points to improve its scientific rigor and impact.:
Comments for authors:
Title:
The title accurately reflects the content of the study and is concise and informative.
Abstract:
1. Even with abstract restrictions the abstract lacks a clear and comprehensive introduction to the topic. It should briefly introduce the significance of hypoxia as a therapeutic target in cancer treatment and provide a context for the research on "Magnetogene" vector development. The importance of cancer-specific gene therapies and the need for targeted delivery systems should be highlighted..
Introduction:
1. The introduction briefly mentions the importance of "Magnetogene" for cancer-specific gene therapy targeting hypoxic tumors but could benefit from a more comprehensive explanation of why this approach is promising compared to other delivery systems. A comparison with existing methods and their limitations would enhance the paper's scientific value.
2. The introduction should conclude with a clear and focused objective statement outlining the specific goals and aims of the study. This will provide a strong foundation for the reader to understand the study's purpose and research questions.
Methods:
1. The description of the pHRE-Luc plasmid (lines 82-88) could be more concise and clear. While it conveys the overall design, some sentences are complex and may be challenging for readers to follow. Simplifying the language and providing a clear summary of the plasmid's components and purpose would improve clarity.
2. In Vitro Studies: The section lacks information on the sample size or the number of replicates performed for the in vitro experiments (lines 94-100). It is essential to include this information to assess the statistical significance and reliability of the results.
3. The synthesis process for MNPs (lines 106-114) is briefly described but may benefit from more explicit steps and parameters. Providing specific reaction conditions and details on the methodological modification compared to the original protocol (Bae et al., 2012) would strengthen the reproducibility of the study.
4. In Vivo Studies: The section does not mention the route and dosage of Magnetogene administration in the mice (lines 133-140). Including these details is crucial for replicability and allows readers to understand how the treatment was applied.
5. Statistical Analysis: While the section mentions performing statistical analysis with GraphPad Prism (lines 150-152), specific statistical tests used and the significance level should be provided.
Results:
1. Clear Presentation: The "Results" section presents the data in a clear and concise manner, with the use of figures to aid in visual interpretation.
2. While the data presented for the in vitro studies are informative, the section lacks specific experimental details, such as the number of replicates and statistical analyses performed. Including this information is crucial for assessing the reliability and significance of the results.
3. Magnitude of Differences: The section should include p-values or other statistical measures to quantify the significance of differences observed in Figure 1 and Figure 5. This will add strength to the conclusions drawn from the data.
Discussion:
1. While the "Discussion" section is well-structured and informative, it would benefit from a concise summary of the study's objectives in the introduction. Additionally, including a brief overview of the methodology used to construct the pHRE-Luc vector and synthesize Magnetogene would provide readers with better context before diving into the discussion.
2. The section could be strengthened by providing a more in-depth comparison of the study's findings with similar studies. For instance, discussing similarities and differences in experimental setups, vector design, and vector efficiency among different research works could add more depth to the discussion.
Conclusions:
1. While the section touches on the challenges and need for continuous evaluation, it would benefit from briefly mentioning some specific findings or key results from the study. This would reinforce the conclusions with evidence from the research.
2. The conclusion can be strengthened by creating a stronger connection to the previous "Discussion" section. For instance, the authors can briefly summarize the main points discussed in the "Discussion" and how they align with the study's overall objective.
Author Response
Response to reviewer 2
The manuscript presents a novel approach for cancer-specific gene therapy using the Magnetogene vector. The authors successfully demonstrate its hypoxic inducibility and targeted delivery in tumor-bearing mice. However, the paper would benefit from addressing the mentioned points to improve its scientific rigor and impact.:
Comments for authors:
Title: The title accurately reflects the content of the study and is concise and informative.
Abstract:
1 Even with abstract restrictions the abstract lacks a clear and comprehensive introduction to the topic. It should briefly introduce the significance of hypoxia as a therapeutic target in cancer treatment and provide a context for the research on "Magnetogene" vector development. The importance of cancer-specific gene therapies and the need for targeted delivery systems should be highlighted.
Response to reviewer 2
The reviewer's suggestion was taken into account. Abstract was modified, giving a clearer introduction and emphasizing the importance of the vector, as well as highlighting how hypoxia is used to provide tumor slectivity and reduce toxicity to healthy tissue.
Introduction:
- The introduction briefly mentions the importance of "Magnetogene" for cancer-specific gene therapy targeting hypoxic tumors but could benefit from a more comprehensive explanation of why this approach is promising compared to other delivery systems. A comparison with existing methods and their limitations would enhance the paper's scientific value.
- The introduction should conclude with a clear and focused objective statement outlining the specific goals and aims of the study. This will provide a strong foundation for the reader to understand the study's purpose and research questions.
Response to reviewer
The reviewer's suggestions were taken into account. The introduction was mostly modified. The points of the use and importance of the Magnetogene vector in cancer gene therapy were highlighted, the main objectives were added and the main purpose of this study was emphasized.
Methods:
- The description of the pHRE-Luc plasmid (lines 82-88) could be more concise and clear. While it conveys the overall design, some sentences are complex and may be challenging for readers to follow. Simplifying the language and providing a clear summary of the plasmid's components and purpose would improve clarity.
Response to reviewer
This section was modified. The description of pHRE-Luc and pGSH1-GFP are more concise and clear. The components and function of both plasmids are explained.
- In Vitro Studies: The section lacks information on the sample size or the number of replicates performed for the in vitro experiments (lines 94-100). It is essential to include this information to assess the statistical significance and reliability of the results.
Response to reviewer
This section was modified. Clearer information was added about the plasmids used and also the information requested by the reviewer.
- The synthesis process for MNPs (lines 106-114) is briefly described but may benefit from more explicit steps and parameters. Providing specific reaction conditions and details on the methodological modification compared to the original protocol (Bae et al., 2012) would strengthen the reproducibility of the study.
Response to reviewer
All the information of the methodology for the synthesis of MNPs is described in the section. The only modification we made was to use another precipitating agent, in this study we used NH4OH and in Bae's work we used NaOH.
- In Vivo Studies: The section does not mention the route and dosage of Magnetogene administration in the mice (lines 133-140). Including these details is crucial for replicability and allows readers to understand how the treatment was applied.
Response to reviewer
Requested information by the reviewer in the "in vivo studies" section has been added and states this:
All Magnetogene nanoparticles vector groups (5000 ng of MNPs, 50µg of pDNA) were injected to the mice through the tail vein, a magnet was placed on the thigh of tumor bearing mice for 1 h (Figure 4)
Complete information on the volumes and concentrations used of the other Magnetogene components can be found in the "Magnetogene synthesis and characterization" section.
- Statistical Analysis: While the section mentions performing statistical analysis with GraphPad Prism (lines 150-152), specific statistical tests used and the significance level should be provided.
Response to reviewer
In the statistical analysis section, it is mentioned that Tukey's test was used for the analysis of all the tests performed in this study (figure 1 and 5).
In the results section (Selective hypoxic expression of pHRE-Luc vector in B16F10 cells) the significance levels of each group used are described and in figure caption of image 1 statistical test used was added.
“Levels of luciferase on pHRE-Luc vector in hypoxic condition 1,615,768 RLUs (p<0.0001) were significantly higher that normoxic condition 379,851 RLUs”
In the “Measurements of Magnetogene levels in mice tissue” section the significance levels between groups were added and in figure caption of image 5 statistical test used was added
Results:
- Clear Presentation: The "Results" section presents the data in a clear and concise manner, with the use of figures to aid in visual interpretation.
- While the data presented for the in vitro studies are informative, the section lacks specific experimental details, such as the number of replicates and statistical analyses performed. Including this information is crucial for assessing the reliability and significance of the results.
Information requested by the reviewer was added. The number of replicates and statistical analysis used were added in the figure captions 1 and 5.
- Magnitude of Differences: The section should include p-values or other statistical measures to quantify the significance of differences observed in Figure 1 and Figure 5. This will add strength to the conclusions drawn from the data.
In the results section (Selective hypoxic expression of pHRE-Luc vector in B16F10 cells) the significance levels of each group used were added
In the “Measurements of Magnetogene levels in mice tissue” section the significance levels between groups were added
Discussion:
- While the "Discussion" section is well-structured and informative, it would benefit from a concise summary of the study's objectives in the introduction. Additionally, including a brief overview of the methodology used to construct the pHRE-Luc vector and synthesize Magnetogene would provide readers with better context before diving into the discussion.
The suggestions proposed by the reviewer were added: a concise summary of the objectives of the study was added in the introduction. The paragraph on the methodology for the construction of the pHRE-Luc vector was modified. We consider the methodology for the synthesis of the Megnetogen to be complete since it contains all the information we used for its synthesis: concentrations, temperature and volumes.
- The section could be strengthened by providing a more in-depth comparison of the study's findings with similar studies. For instance, discussing similarities and differences in experimental setups, vector design, and vector efficiency among different research works could add more depth to the discussion.
Most of the results of this work are compared with other studies in a general way, we believe that this comparison shows the importance and difference of the results obtained.
Conclusions:
- While the section touches on the challenges and need for continuous evaluation, it would benefit from briefly mentioning some specific findings or key results from the study. This would reinforce the conclusions with evidence from the research.
- The conclusion can be strengthened by creating a stronger connection to the previous "Discussion" section. For instance, the authors can briefly summarize the main points discussed in the "Discussion" and how they align with the study's overall objective.
The reviewer's suggestions were considered. The conclusions section was modified, and specific and concise conclusions were added.

Round 2
Reviewer 1 Report
All the comments are addressed
English is fine